# Identification of New Genes and Genetic Variant Loci Associated with Breast Muscle Development in the Mini-Cobb F2 Chicken Population Using a Genome-Wide Association Study

**DOI:** 10.3390/genes13112153

**Published:** 2022-11-18

**Authors:** Yang He, Hongmei Shi, Zijian Li, Jiajia Kang, Mengyuan Li, Mengqian Liu, Yong Liu, Jinbo Zhao, Tengfei Dou, Junjing Jia, Yong Duan, Kun Wang, Changrong Ge

**Affiliations:** 1College of Animal Science and Technology, Yunnan Agricultural University, Kunming 650201, China; 2Kunming Animal Health Supervision, 118 Gulou Road, Kunming 650000, China

**Keywords:** chicken, breast muscle development, genome-wide association study, candidate genes

## Abstract

Native chicken has become a favorite choice for consumers in many Asian countries recently, not only for its potential nutritional value but also for its deep ties to local food culture. However, low growth performance and limited meat production restrict their economic potential. Conducting a genome-wide association study (GWAS) for chicken-breast muscle development will help identify loci or candidate genes for different traits and potentially provide new insight into this phenotype in chickens and other species. To improve native chicken growth performance, especially breast muscle development, we performed a GWAS to explore the potential genetic mechanisms of breast muscle development in an F2 population constructed by reciprocal crosses between a fast-growing broiler chicken (Cobb500) and a slow-growing native chicken (Daweishan mini chicken). The results showed that 11 SNPs, which exceeded the 10% genome significance level (*p* = 1.79 × 10^−8^) were considered associated with breast muscle development traits, where six SNPS, NC_006126.5: g.3138376T>G, NC_006126.5: g.3138452A>G, NC_006088.5: g.73837197A>G, NC_006088.5: g.159574275A>G, NC_006089.5: g.80832197A>G, and NC_006127.5: g.48759869G>T was first identified in this study. In total, 13 genes near the SNPs were chosen as candidate genes, and none of them had previously been studied for their role in breast muscle development. After grouping the F2 population according to partial SNPs, significant differences in breast muscle weight were found among different genotypes (*p* < 0.05), and the expression levels of *ALOX5AP*, *USPL1*, *CHRNA9*, and *EFNA5* among candidate genes were also significantly different (*p* < 0.05). The results of this study will contribute to the future exploration of the potential genetic mechanisms of breast muscle development in domestic chickens and also support the expansion of the market for native chicken in the world.

## 1. Introduction

Chicken meat is one of the major meat products in the world and is also a cheap protein source. According to the World Food and Agriculture-Statistical Yearbook 2021 (www.fao.org, updated on 28 October 2022), chicken meat products accounted for 35% of the world’s meat products in 2019, while only 25% in 2000. The rapid increase in the data is closely related to the selection of the breast muscle in chickens [1]. However, not all consumers are willing to pay for this fast-growing chicken. Furthermore, in some Asian countries, consumers are more interested in slow-growing native chickens due to their greater chewiness and nutrient richness [2,3]. Moreover, some native chickens have become a favorite choice for consumers in poor regions because they can provide both essential meat and eggs to locals with a high level of environmental and disease resilience. Nevertheless, these native chickens often are small in size, have poor growth, and weak breast muscles, which dramatically restrict their meat production and economic value. As a result, improving their growth performance and meat production has become a hot topic at present.

In order to improve the meat production of native chickens effectively, it is necessary to explore the potential genetic mechanisms of breast muscle development. Genome-wide association analysis reveals the potential genetic mechanisms of traits at a DNA level, and the method is driven by the association of genomic variation (mainly single nucleotide polymorphisms (SNPs)) and traits [4]. This technology has been widely used to explore the mechanisms of complex traits, including human diseases [5,6,7], plant development or disease resistance [8,9], as well as animal growth and reproductive capacity [10,11] in past years. Similarly, a GWAS has also been applied to the study of chicken development, especially breast muscle development and some significant candidate genes or genomic regions have been identified [12,13,14]. Interestingly, genetic variants within the genome significantly associated with breast muscle development appear to be diverse and are often associated with other traits, such as body weight, full evisceration weight, and so on [15]. In some early studies, these traits have been shown to be significantly correlated with breast muscle weight in chickens [16]. So does genetic variation in these traits, which are significantly associated with breast muscle weight, also affect breast muscle development? Unfortunately, there are few studies involved in this area of research currently.

With the proposal and use of Pam genomics, the annotation of functional regions in the domestic chicken genome is more precise, and this was achieved with the contribution of some native chickens. A previous report used 20 domestic chicken species distributed worldwide to construct a Pam genome and identify 1335 and 3011 coding genes and noncoding RNAs never found in the chicken 6th edition genome (GRCg6a), in which the Daweishan mini chicken, a native chicken distributed in Yunnan, China, plays a key role [17]. The Daweishan mini chicken is one of the most valuable native chickens. It is mainly distributed in the mountainous areas of tropical and subtropical western China, although its growth performance, weight, and size are much smaller than other breeds in China [18]. Our previous study found that compared to other native chickens, the Daweishan mini chicken’s cellular metabolism was more active in the breast muscle, while the expression levels of genes related to muscle development were lower [19]. Unexpectedly, the gene Myostatin (MSTN), which negatively regulates muscle growth, plays a dual role in promoting and repressing growth throughout the muscle development of Daweishan mini chickens, suggesting a unique pattern of muscle development in this chicken [20].

To explore the potential genetic mechanisms of breast muscle development in domestic chickens and improve the status of breast muscle development in native chickens, here, we constructed a F2 population using reciprocal crosses between a fast-growing broiler chicken (Cobb500) and a slow-growing native chicken (Daweishan mini chicken). We identified the SNPs within the genome of the F2 population using whole genome sequencing and conducted a GWAS by combining traits related to breast muscle development. Finally, a preliminary validation was performed for the results of the post-GWAS analysis to improve the credibility of our results.

## 2. Materials and Methods

### 2.1. Animal Experimentation Ethical Statement

All animal experimental procedures were approved and guided by the Yunnan Agricultural University Animal Care and Use Committee (approval ID: YAUACUC01, publication date, 10 July 2013).

### 2.2. Animals and Management

The crossbreed design for this experiment was applied to a reciprocal crossing between fast-growing broiler chickens (Cobb500) and a slow-growing native chickens (Daweishan mini chicken). Cobb500 (CC) and Daweishan mini chickens (MM) were reared in individual cages and fed a standard diet (19% CP; 2840 kcal/kg ME; 3.5% Ca and 0.32% nonphytate P) at the Yunnan Agricultural University’s experimental chicken farm. Crossbreeding experiments were used to produce the F1 population by artificial insemination as a 1:1 ratio of pairs, and to construct twenty families through the mating of MM ♂ × CC ♀ or CC ♂ × MM ♀. Every F1 male from the twenty populations was mated with another two females to produce a total of 614 F2 chickens in 60 half-families.

A total of 614 healthy chicks were obtained and transferred to the brooder house after being vaccinated. The rearing density was adjusted with time, and the density was 24–28 birds/m^2^ before 4 weeks and adjusted to 15–20 birds/m^2^ from 4 to 7 weeks. In the eighth week, the chickens were transferred to three-tier cages to rear until 20 weeks, and the density was 4–6 birds/m^2^. The light pattern was 23 h for the first 3 days of the first week, then decreased weekly until 10 h in the seventh week. The dietary standards are listed in Table 1.

### 2.3. Phenotypes

At 90 days, 478 healthy chickens were chosen from the F2 population for phenotype testing. Before slaughter, 4 mL of blood was extracted from the chicken wing vein into a tube and stored at 4 °C for the DNA extract. The birds were weighted and electrically stunned in a water bath (240 mA, 120 V, 5 s), then killed by neck cut. The carcasses were cooled in a chilling room (4 °C) for 45 min before the phenotypes were evaluated (see Table 2).

### 2.4. Whole Genome Sequencing and Quality Control

The chicken blood used for total genome DNA was extracted using a Tiangen kit (Beijing, China), and the DNA concentration as well as purity was measured using a Nanodrop^TM^ 1000 spectrophotometer (Thermo Scientifc, Waltham, MA, USA) and electrophoresis. The genomic DNA was randomly broken into 350 bp fragments and was treated by the TruSeq library construction kit (Illumina, Santiago, Californian, USA) for constructing the whole genome library. Sequencing was conducted by the Illumina HiSeq 4000 (Illumina, Inc.) with strategies of PE150 at a depth of 10×. The raw data obtained from sequencing were cleaned by removing the contained paired-end reads and low-quality reads using BWA software (version: 0.7.8-r455) [21]. The clean data were aligned with the domestic chicken reference genome GCF_000002315.6_GRCg6a. The software SAMtools (version: 0.1.19-44428cd) was used to remove duplicates from the aligned results, then detect the SNPs within the genome and filter low-quality SNPs [22]. Finally, the software ANNOVAR was used to obtain SNP information using region-based annotations [23].

### 2.5. Genome-Wide Association Study

Highly consistent SNPs were obtained after being filtered with minor allele frequency > 0.01 and missing < 0.2 using software PLINK (version: 1.9) [24], and a total 5580468 SNPs were used for the GWAS. GEMMA software [25] was used for correlation between SNPs and traits based on the mixed linear model (MLM), and the formula was follows:y = Wa + Xb + Zc + ε
where y represented the phenotypes, X was the genotypes, the a (fixed effect) was the matrix of population structure calculated using ADMIXTURE software [26], and the c was the matrix of kinship relationship calculated using GCTA software [27], Wa and Xb were fixed effects, Zc were random effects, and ε was residual. The GWAS threshold was determined by the Bonferroni correction methods, and 10% (*p* = 1.79 × 10^−8^), 5% (*p* = 8.96 × 10^−9^), as well as 1% (*p* = 1.79 × 10^−9^) genome-wide significance levels were set. Considering the complex of traits, SNPs that exceed the 10% genome-wide significance level were treated as candidates.

### 2.6. SNP Identification, Candidate Gene Annotation, and QTL Overlapping

Genes located within 100 kb upstream and downstream of the significant SNPs were selected as candidates based on the methods in reference [28]. GO annotation of candidate genes was then performed using the Gene Ontology Consortium (http://geneontology.org, updated on 28 October 2022). The chicken quantitative trait locus (QTL) database (Chicken QTLdb, https://www.animalgenome.org/cgi-bin/QTLdb/GG/index, updated on 28 October 2022) was searched based on SNP positions to determine whether SNPs had been previously reported in QTLs.

### 2.7. mRNA Expression Analysis

Four candidate genes were validated by real-time polymerase chain reactions. The total RNA was extracted from the breast muscle of different genotypes and converted by reverse transcriptase to cDNA (Takara, Kusatsu, Japan). We used a SYBR premix Ex-Taq TM II (Takara, Kusatsu, Japan) in an ABI 7500 fast real-time PCR system (Thermo Scientifc, Waltham, MA, USA) to perform real-time PCR. The 2^−∆∆CT^ method was used to determine relative expression, and *GADPH* was used as the internal control for normalization. Every sample was repeated in triplicate and primer sequences are listed shown in Table 3.

### 2.8. Statistical Analysis

All data were analyzed by SPSS statistical software (version 18.0, SPSS, Chicago, IL, USA) at a 5% significance level after a normal distribution test. The correlation between the breast muscle development-related traits was calculated by Spearman’s rank correlation coefficient. The breast muscle weight and the expression levels of candidate genes in different genotypes were compared by the independent sample *t*-test. R software was used for the visualization of all the data.

## 3. Results

### 3.1. Breast Muscle Development-Related Traits in the F2 Population and Correlation Analysis

We recorded and calculated the values of breast muscle development-related traits, including body weight, breast muscle weight, breast width, breast depth, keel length, full net bore weight, and breast muscle percent for 478 F2 individuals at 90 days, and the results are shown in Table 4.

Among these traits, the lowest coefficients of variation were found for breast width, breast depth, and keel length, while the highest were for breast muscle weight and sternum weight. In addition, the maximum and minimum values of individual traits differed significantly, such as body weight, where the difference was about 2400 g. Considering that these traits are often thought to be strongly correlated with breast muscle development, we also examined the relationship between these traits and breast muscle weight in the F2 population. Depending on the situation, Spearman’s rank coefficient or Pearson’s correlation coefficient was used to describe the correlation between variables. We first tested whether these traits followed a normal distribution and found that except for keel length, all other traits (including breast muscle weight) did not (see Figure 1), so we chose Spearman’s rank correlation coefficient to continue to analyze the correlation between them.

The results are shown in Figure 2. The correlation between body weight and breast muscle weight was the most significant (correlation = 0.79, *p* < 0.001), while sternum weight and full evisceration weight were less correlated with breast muscle weight (0.39 and 0.36, *p* < 0.001). Additionally, sternum weight was negatively correlated with breast percent (correlation = −0.21, *p* < 0.001). Based on the results, we selected all traits but sternum weight and full evisceration weight to continue the subsequent analysis.

### 3.2. Genetic Variation in the F2 Population

All samples were sequenced and produced a total of 5619.344 G of raw data and 5511.828 G of clean data after quality control, of which the average Q20 was 96.31%, Q30 was 91.1%, and GC content was 41.73%. The results after comparing with the reference genome of chicken (GCF_000002315.6_GRCg6a) showed that the average sequencing coverage depth was between 7.32× and 9.83×, while the 1× coverage was above 96.41%, indicating that the sequencing results were normal and could be used for subsequent variant analysis. As shown in Figure 3A, four common variants were found within the F2 population genome, including SNP, Indel, SV, and CNV, and they were distributed across almost all chromosomes within the entire sample genome. In total, 2,308,088,443 SNPs were identified, with an average of 4,828,636.91 SNPs per sample detected. However, after filtering (missing < 0.2, MAF > 0.01) we obtained a total of 5,597,890 SNPs. As shown in Figure 3B, the majority of SNPs (5,580,468) were distributed among the 33 autosomes and two sex chromosomes (“W” and “Z”) in chicken. However, some of the SNPs (17,422) were located in microchromosomes in the chicken genome that has not been fully annotated, and we excluded these SNPs in the subsequent analysis because of the limited knowledge of these regions.

### 3.3. Candidate SNPs and Genes

Since the population structure may have an effect on the GWAS results, we plotted the quantile–quantile plots (Q–Q plot) for all traits. As shown in Figure 4, the observed *p*-values largely overlapped with the predicted *p*-values, and the genome inflation coefficients were distributed between 0.983 and 1.02, which implied that the population structure was effectively corrected by using MLM and the result was trustworthy. Subsequently, we plotted Manhattan plots for all traits based on the GWAS results and screened SNPs significantly associated with traits using the 10% significance level of the genome as the threshold after Bonferroni correction (Figure 4).

As shown in Table 5, a total of 11 SNPs exceeded the 10% genomic significance level and were significantly associated with five traits, while the SNPs associated with breast width failed to reach the minimum threshold level. Eleven SNPs were mainly distributed on chicken chromosomes 1, 2, 4, W, and Z, while the phenotypic variance explained by these SNPs ranged from 5.32 to 24.29%. The SNP (rs731233898) located at 176412883 bp on chromosome 1 was significantly associated with breast muscle weight (*p*-value = 6.42 × 10^−10^) as well as body weight (*p*-value = 2.47 × 10^−10^), and its nearby genes *ALOX5AP* (downstream 3688 bp) and *Uspl1* (downstream 16,919 bp) emerged as potential candidate genes for breast muscle weight and body weight. Several SNPs (rs733077910, rs738075929, and rs315941028) were found to be significantly associated with traits such as breast percent, breast depth, body weight, and keel length, and genes *KLHL1* and *PCDH9* are potential candidate genes for these traits. Two known SNPs (rs313037222 and rs313037222) were significantly associated with the keel length trait alone, and their surrounding genes, *Rbm47*, *CHRNA9*, as well as *COG6*, became candidates for keel length. Furthermore, there were six new SNPs identified in this study (NC_006126.5: g.3138376T>G, NC_006126.5: g.3138452A>G, NC_006088.5: g.73837197A>G, NC_006088.5: g.159574275A>G, NC_006089.5: g.80832197A>G, and NC_006127.5: g.48759869G>T). These SNPs were significantly associated with multiple traits, with the SNP located on chromosome Z (NC_006127.5: g.48759869G>T) being significantly associated with keel length only. These newly identified genes around SNPs include *SUN3* (LOC428505), *NEDD4L*, *ERVK-8*, and *KCNA1*, thus becoming potential candidates for related traits.

### 3.4. Comparing with Previous QTL

We searched for the region in which these 11 SNPs were located (+/−100 kb) in the chicken QTL database, to compare whether they overlapped with previously reported QTLs related to breast muscle development. It was found that three known SNPs (rs733077910, rs731233898, and rs315941028) and two newly identified SNPs (NC_006089.5: g.80832197A>G, NC_006088.5: g.73837197A>G) overlapped with four previously reported QTLs related to body weight (QTLs: 7001, 9752, 1858, and 12461). Three SNPs (rs733077910, rs315941028, and NC_006088.5: g.73837197A>G) overlapped with QTLs (QTL:9301, QTL:16714, and QTL:9298) associated with breast depth. However, SNPs associated with other traits did not overlap with their corresponding QTL, such as those associated with the keel length trait (rs313037222, rs13973830) or with the breast muscle weight trait (rs731233898), and whether the regions in which these SNPs are located are potential QTLs affecting these traits still needs further discussion. Interestingly, we found that three SNPs (rs313037222, NC_006089.5: g.80832197A>G, and NC_006088.5: g.73837197A>G) with previously reported QTLs for breast muscle weight (QTL: 9422, QTL: 13386, and QTL: 12462) overlapped, indicating that these SNPs are correlated with breast muscle weight and confirming the reliability of the traits we selected for correlation with breast muscle development.

### 3.5. GO Annotation

We performed GO annotation of the 13 candidate genes obtained and analyzed the association between their involved functions and biological processes. However, only two genes, CHRNA9 and EFNA5, were successfully annotated (Figure 5). CHRNA9, together with EFNA5, are involved in a variety of biological processes, including responses to external stimuli (GO: 0009605), cell surface receptor signaling pathways (GO: 0007166), anatomical structure morphogenesis (GO: 0009653), and animal organ development (GO: 0048513). The cellular components of both of them were intrinsic components of the plasma membrane (GO: 031226). Moreover, the molecular function of CHRNA9 was excitatory extracellular ligand-gated ion channel activity (GO: 0005231), acetylcholine-gated cation-selective channel activity (GO: 0022848), calcium channel activity (GO: 0005262), and calcium ion transmembrane transporter activity (GO: 0015085). Unfortunately, the genes *ALOX5AP*, *KLHL1*, *PCDH9*, *NEDD4L*, *ERVK-8*, *Uspl1*, *KCNA1*, *COG6*, *Rbm47*, *gag-pro*, and *Sun3* were not annotated.

### 3.6. The Relationship between Breast Muscle Weight and the Expression Levels of Candidate Genes

The breast muscle weight of different genotypes and corresponding candidate gene expression levels are shown in Figure 6. Most of the cDNA sequences and protein structures of the above candidate genes were obtained by computer simulation prediction, and only a small number of genes (*CHRNA9*, *EFNA5*, *ALOX5AP*, and *USPL1*) were confirmed. Therefore, we used the SNPs corresponding to these four genes (rs313037222, rs731233898, and NC_006127.5: g.48759869G>T) to divide the F2 population into different genotypes and compare their breast muscle weights as well as the expression levels of these candidate genes within the breast muscle. The results showed that rs731233898 divided the F2 population into three genotypes (GG, GC, and CC), in which the breast muscle weight of the GG type was significantly smaller than that of the GC (*p* < 0.0001) and CC (*p* < 0.01) types, and the genes corresponding to this SNP, *ALOX5AP* and *Uspl1*, also showed the same expression level trend. The expression level of *ALOX5AP* in the GG type was significantly smaller than that in the GC and CC types (*p* < 0.01), while the expression level of *Uspl1* in breast muscle was also significantly larger in the mutant type than in the wild type (*p* < 0.05). rs313037222 also classified the F2 population into three genotypes (CC, CT, and TT), where the CC type (wild type) had a very significantly lower breast muscle weight than the CT type (*p* < 0.0001) and the expression level of *CHRNA9* in the CC type breast muscle was significantly lower than in the CT and TT types (*p* < 0.01). Due to the small sample size of the TT type, we did not compare it with the other two genotypes. Finally, our newly identified SNP (NC_006127.5: g.48759869G>T) distinguished the F2 generation into two genotypes (GG and GT), while the breast muscle weight of the GG type was extremely significantly smaller than that of the GT type (*p* < 0.0001). Interestingly, the expression level of EFNA5 was significantly higher in GG type breast muscle than in GT type (*p* < 0.01), implying that the expression level of EFNA5 was negatively correlated with breast muscle weight.

## 4. Discussion

### 4.1. Traits and Correlation Coefficients

We have noted that some traits have large coefficients of variation (such as sternum weight and breast muscle weight), which reflect large genetic differences between individuals in the F2 populations, and we attribute this to a combination of hereditary and nonhereditary factors. The Daweishan mini chickens are an almost unselected native breed with underdeveloped breast muscles and small breast bones [19]. In other words, there is a considerable potential for genetic improvement. When it is crossed with commercial broilers (a breed selected for pectoral muscle), the offspring will have large differences in breast muscle weight and sternum weight, resulting in considerable coefficients of variation. Moreover, nonhereditary factors can also affect the coefficient of variation, such as an individual’s diet, age, and trait measurement process [29]. Since errors are inevitable in the process of measurement for chicken-breast muscle weight and sternum weight, this leads to actual values that are inaccurate in some individuals and eventually affects the coefficient of variation. In addition to standardized operation, an effective solution includes the use of in vivo ultrasound [30] or computer technology to establish a deep learning algorithm for detection [31] to improve measurement efficiency while reducing operational errors.

The degree of correlation between variables is usually reflected using correlation coefficients, among which Spearman’s rank coefficient and Pearson’s correlation coefficient are commonly used. The Pearson’s correlation coefficient is able to assess the linearity of the relationship between continuous variables, while the Spearman’s correlation coefficient evaluates their monotonic relationship [32]. In most studies, the Pearson’s correlation coefficient is usually used to describe the correlation between traits [28,33], but in this study, the relationship between these traits, such as breast muscle weight and breast depth (Figure 1), did not show complete linearity. Although they showed the same trend, they did not have a stable rate of change, which is more consistent with monotony. So, using Spearman’s rank coefficient, we can describe their relationships more accurately. Secondly, outliers in the variables often affect the Pearson’s correlation coefficient results; in contrast, the Spearman‘s correlation coefficient has a stronger tolerance for outliers in the variables [34]. Lastly, the Pearson’s correlation coefficient usually requires that all variables follow the normal distribution (the Gaussian distribution) [35]. However, only the keel length followed normal distribution in this study, while the others did not. Hence, the nonparametric Spearman’s coefficient is more appropriate because it is not required for the distribution relationship of the variables.

### 4.2. Selecting of Candidate SNP

We obtained a total of 5,597,880 SNPs by whole-genome sequencing of the F2 population. However, we did not apply all SNPs to the GWAS but only considered those distributed within known chromosomal regions (5,580,468). SNPs are distributed on almost all the chromosomes of chickens, including large chromosomes and microchromosomes. Microchromosomes have higher GC and CpG content and less gene content than large chromosomes, but we have not been able to fully understand the composition of these microchromosomes due to the limited coverage during sequencing [36]. In the present study, we also obtained 17,422 SNPs distributed in these microchromosomes, which we did not include in the subsequent GWAS analysis because of the very limited knowledge of these regions. In addition, we used the domestic chicken reference genome (GRCg6a), which contains 33 autosomes and two sex chromosomes and is missing some chromosomal information (including GGa34-39) compared to the seventh version of the domestic chicken reference genome, which may lead to some missing genetic variants. However, at the same time, the seventh edition of the domestic chicken reference genome was assembled from the broiler genome, while the sixth edition was from the red jungle fowl. Therefore, we believe that using the sixth genome as a reference can uncover more information on genetic variation from the F2 population.

After the GWAS, we used SNPs with a p-value exceeding the 10% genomic significance level, after the Bonferroni correction, as candidate genes. While in some other GWAS studies in chickens, the SNP significance threshold is usually set to exceed the 5% genomic significance level [37,38]. The Bonferroni correction is very effective for rejecting false positive loci in an association analysis, especially for samples of small sizes. However, when faced with a large number of samples, such as millions of SNPs obtained from whole genome sequencing, the Bonferroni correction appears to be very conservative because it does not consider the linkage disequilibrium between these SNPs, often resulting in a large number of false negative results [39]. Additionally, all traits analyzed by the GWAS in this study were quantitative traits, and quantitative traits, such as weight, height, and even disease occurrence, are controlled synergistically by multiple microeffect genes. Appropriately lowering the threshold level facilitates the screening of these microeffect loci, so in this study we set a 10% genome significant level as the threshold and eventually identified a SNP (rs313037222) that overlapped with the breast muscle weight QTL (QTL: 9422). Nevertheless, even though we relaxed the threshold level, we still failed to obtain SNPs significantly associated with breast width. Combined with the correlation analysis between breast width and breast muscle weight (Figure 2, correlation coefficient = 0.61 ***), the possibility of false negatives in the analysis cannot be ruled out, and further validation is still needed.

### 4.3. Candidate Genes

We selected genes in their neighborhoods as potential candidates for affecting breast muscle development based on the location of the 11 SNPs within the genome (Table 5). We first performed GO annotation on these candidate genes, however, only two genes, *CHRNA9* and *EFNA5*, were successfully annotated (Figure 5). Therefore, we searched the information on these nine genes and found that the functional studies of these genes in chickens are still at a very preliminary stage, with only a few genes (*CHRNA9*, *EFNA5*, *ALOX5AP*, and *USPL1*) having their coding regions and proteins confirmed, and most of them are still predicted.

A total of six SNPs (rs733077910, rs738075929, rs731233898, rs13973830, NC_006088.5: g.73837197A>G, NC_006088.5: g.159574275A >G) were distributed throughout the range of chromosome 1 in domestic chickens. These SNPs were significantly associated with traits such as breast muscle weight, body weight, and keel length, and they were concentrated in the region 159,570,000 to 176,410,000 bp on chromosome 1, indicating that key genes affecting these traits were distributed in this region. Arachidonate 5-lipoxygenase-activating protein (*ALOX5AP*) is a protein-encoding gene that is located in the plasma membrane and participates in leukotriene synthesis in combination with 5-lipoxygenase. This gene has been shown to be involved in the development of a variety of human diseases, including type II diabetic nephropathy [40], Lupus Erythematosus [41], and interstitial lung disease [42]. Although the ALOX5AP protein has been found to be highly conserved in several species, including chickens, there are no reports on the functional role of *ALOX5AP* in domestic chickens. Interestingly, one report has indicated that *ALXO5AP* was also involved in the fat metabolism pathway in mice and was negatively correlated with obesity [43]. This report made us consider the increasing incidence of myopathy in fast-growing broilers, especially the white stripe. The cause of this disease is still unknown, but it is generally believed to be related to hypoxia in the breast muscle [44]. In addition, the disease causes an increase in breast muscle weight and intramuscular fat content and produces continuous white streaks on the breast muscle, which affects and impairs the overall quality of the broiler breast muscle [45]. When we genotyped the F2 population according to SNPs and examined their breast muscle weight and *ALOX5AP* expression levels, we found a positive correlation between them. In other words, the higher the breast muscle weight, the higher the expression level of *ALOX5AP*. The relationship between the phenomenon and SNPs is not yet clear and deserves further investigation. Ubiquitin specific peptidase-like 1 (*USPL1*) is a SUMO peptidase with noncatalytic functions and does not bind or cleave ubiquitin [46]. It is mainly distributed within the Cajal body in cells and plays an important role in RNAPII-mediated snRNA transcription [47], but there are no reports of its function in chickens. Kelch-like family member 1 (*KLHL1*) is a protein-coding gene that belongs to the actin family, which is associated with cellular excitation [48], but few studies have addressed the role of *KLHL1* in chickens. Protocadherin 9 (*PCDH9*) encodes a member of the protocadherin family, and cadherin superfamily, of transmembrane proteins containing cadherin domains. In some human diseases, *PCDH9* is negatively correlated with the proliferation and apoptosis of tumor cells [49] and is also a target of some anticancer molecules [50]. In mouse nerves, *PCDH9* is involved in the formation of specific neural circuits [51], while in chickens, only a specific spatiotemporal expression pattern of *PCDH9* was found in chick’s embryonic spinal cords, suggesting that it can affect the development of the spinal cord and its surrounding tissues [52]. A component of oligomeric golgi complex 6 (*COG6*) encodes a subunit of the conserved oligomeric Golgi complex that is required for maintaining normal structure and activity of the Golgi apparatus. In mammals, this gene is associated with male gonad development [53], but its function in chickens is not known. Potassium voltage-gated channel subfamily A member 1 (*KCNA1*) is a coding-gene and mutations in this gene have been linked to myokymia with periodic ataxia (AEMK) in humans [54]. Moreover, in chickens, this gene was associated with sperm motility in males [55].

The SNP, NC_006089.5: g.80832197A>G, located on chromosome 2 at 80,832,197 bp, was found to be significantly related to several traits, including body weight, keel length, and breast muscle percent, and the gene close to it is *SUN3*. Sun domain-containing protein 3-like (*SUN3*), currently named LOC428505 in the chicken reference genome (GRCg6a and GRCg7b), is an encoding gene but its biological function is still mainly predicted. Recent studies found that *SUN3* plays an important role in the plasticity of mouse sperm heads [56]. When *SUN3* is knocked out in male mice using cas9 technology, it leads to a dramatic reduction in sperm count and the development of spherical spermatozoa in male mice, which are associated with sperm head abnormalities. *SUN3* is also overexpressed in arthritis and is a potential marker molecule for arthritis [57]. Interestingly, *SUN3* also seems to play a key role in plant development and reproduction [58], and it was found that *SUN3* exhibits a specific spatiotemporal expression pattern in plants and is regulated by plant development [59]. At present, there are no relevant studies on the function of the *SUN3* gene in chickens, while the GWAS results of the present study showed that *SUN3* was strongly correlated with the keel length and overlapped with the previously reported QTL of breast muscle weight. Therefore, we believe that the relationship between *SUN3* and chicken-breast muscle is deserving of indepth study.

A total of four SNPs, rs315941028, NC_006126.5: g.3138376T>G, NC_006126.5: g.3138452A>G, NC_006127.5: g.48759869G>T, were distributed across the two sex chromosomes (W and Z) of chicken, and four candidate genes were obtained. Ephrin A5 (*EFNA5*) is a member of the ephrin gene family, prevents axon bundling in cocultures of cortical neurons with astrocytes, and a model of late-stage nervous system development and differentiation. Early studies found that *EFNA5* was able to inhibit the growth of motor neurons adjacent to it in a direct or indirect manner [60]. Recent studies have shown that interactions between motor neurons and muscle stem cells contribute to the establishment of functional muscle during development and regeneration [61], suggesting that *EFNA5* is a potential regulator of muscle development. Unfortunately, there is no research on chicken-breast muscle development associated with this gene, and in this study, the mutation of SNP (NC_006127.5: g.48759869G>T) leads to a decrease in the expression level of *EFNA5* in heavy breast muscle. Furthermore, there were only 20 chickens that had mutations at this locus in the whole F2 population, while the small sample size was not enough for us to make an accurate judgment of its potential function, and further validation is still needed. NEDD4-like E3 ubiquitin protein ligase (*NEDD4L*) encodes a member of the Nedd4 family of HECT domain E3 ubiquitin ligases. In humans, *NEDD4L* is involved in the process of several diseases, including the inhibition of hepatocellular carcinoma cell growth [62] and involvement in viral replication [63], while its mutation is associated with hypertension in chronic kidney disease [64]. Currently, there are no functional studies on this gene in chickens. *Gag-pro* and *ERKV-8* are components of genomic endogenous retroviruses, and studies related to their association with growth and development in domestic chickens are scarce, and it is not clear how they relate to breast muscle development.

## 5. Conclusions

In conclusion, we performed a GWAS to explore the potential genetic mechanisms of breast muscle development in an F2 population constructed by reciprocal crosses between a fast-growing broiler chicken (Cobb500) and a slow-growing native chicken (Daweishan mini chicken). Our results showed that the coefficient of variation among breast muscle weight and sternum weight was the highest of all the traits related to breast muscle development, but their correlations were weaker. The GWAS revealed that 11 SNPs significantly associated with breast muscle development, six of which were previously unknown, and three overlapped with previous QTLs for breast muscle development. Based on the distribution of these SNPs across chicken genomes, we identified 13 candidate genes, of which the expression levels of *ALOX5AP*, *USPL1*, *CHRNA9*, and *EFNA5* were correlated with breast muscle weight, suggesting they might indirectly or directly regulate breast muscle development. Our results will contribute to future investigations and studies on the potential genetic mechanisms of breast muscle development in domestic chickens.

## Figures and Tables

**Figure 1 genes-13-02153-f001:**
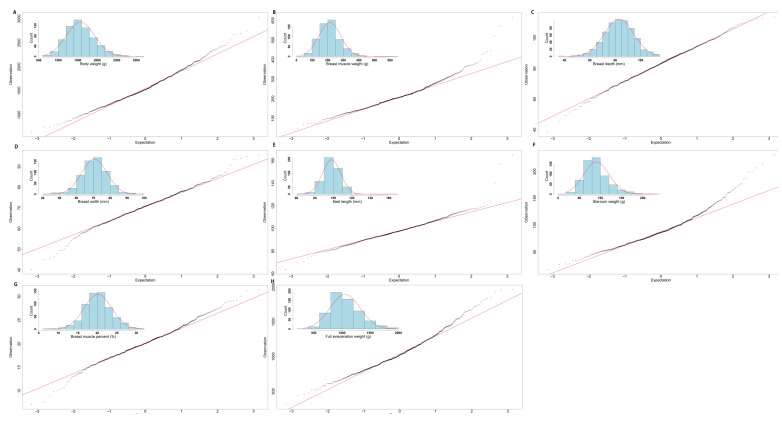
The normal distribution test for breast muscle development-related traits. (**A**) Body weight (g); (**B**) breast muscle weight (g); (**C**) breast depth (mm); (**D**) breast width (mm); (**E**) keel length (mm); (**F**) sternum weight (g); (**G**) breast muscle percentage (%); and (**H**) full evisceration weight (g).

**Figure 2 genes-13-02153-f002:**
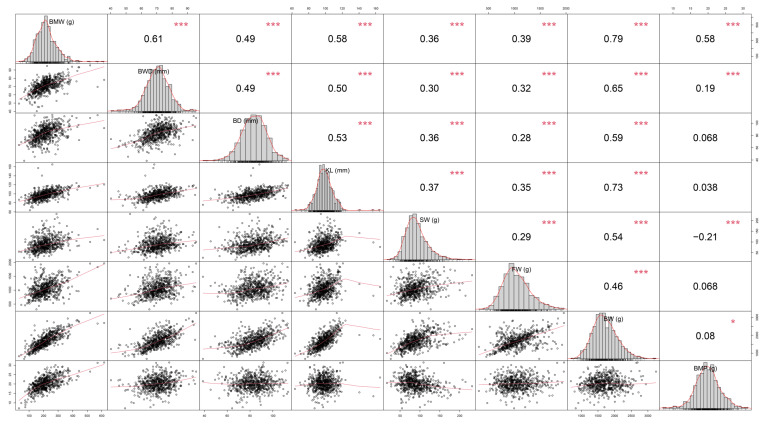
Spearman’s rank correlation coefficient between breast weight and other traits. (**BW**) Body weight (g); (**BMW**) breast muscle weight (g); (**BD**) breast depth (mm); (**BWD**) breast width (mm); (**KL**) keel length (mm); (**SW**) sternum weight (g); (**BMP**) breast muscle percentage (%); and (**H**) full evisceration weight (g); (**FW**) full evisceration weight (g).Statistically significant differences are indicated by * (*p* < 0.05), *** (*p* < 0.001).

**Figure 3 genes-13-02153-f003:**
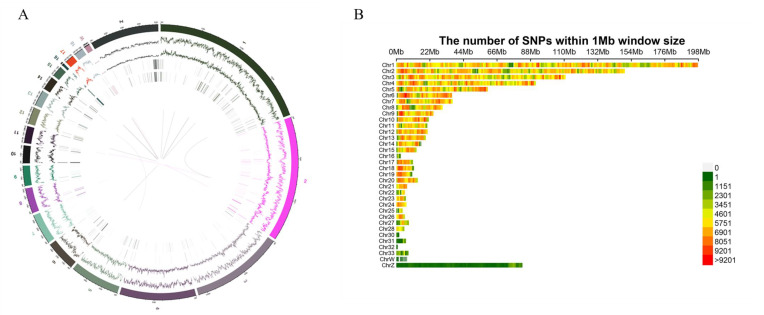
The distribution of variation in F2 populations (**A**) The distribution of variants within autosomal and sex chromosome of the chicken genome, from the outside to the inside of the circle, was SNP, Indel, SV, and CNV. (**B**) The distribution of SNPs within the 1 Mb window on autosomes and sex chromosomes of the chicken genome.

**Figure 4 genes-13-02153-f004:**
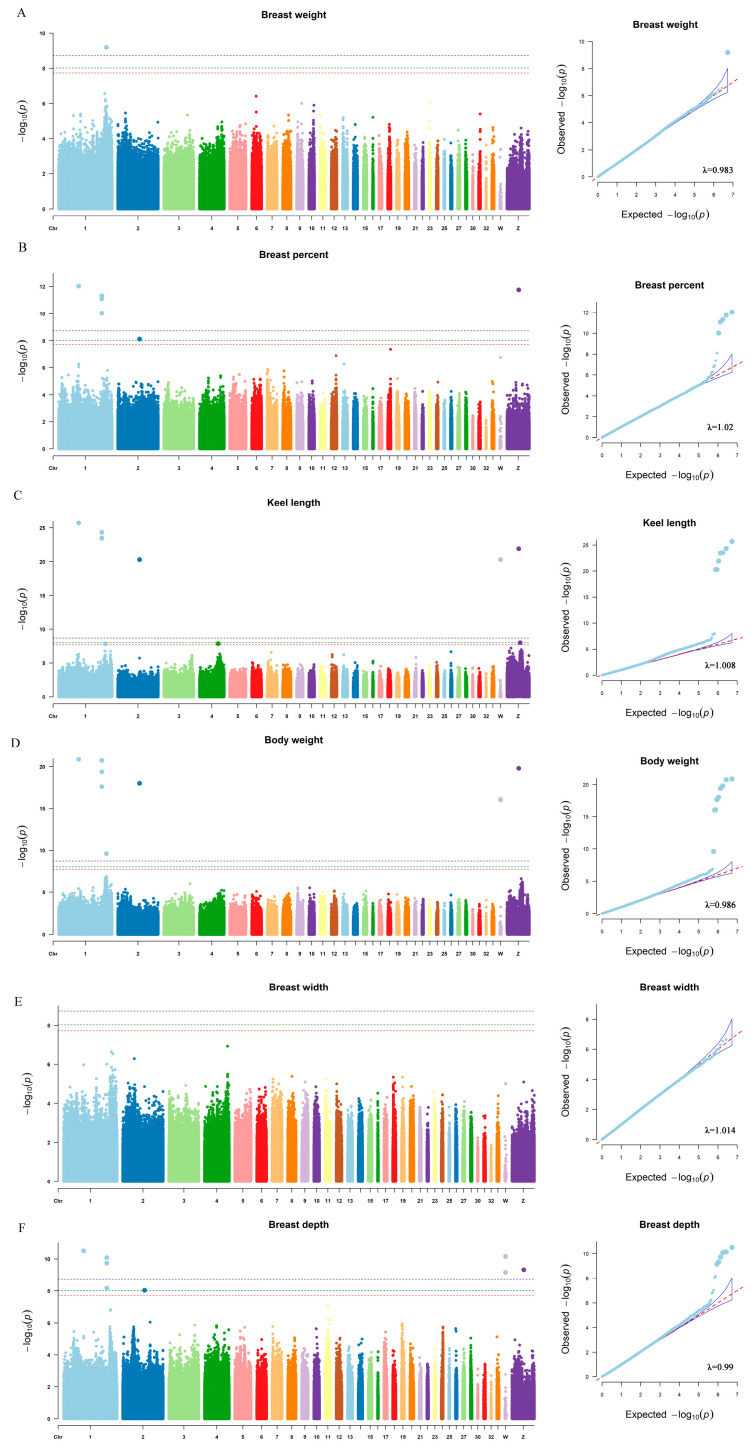
The Manhattan plot and Q–Q plot for breast weight and other traits. In the Manhattan plot, the X axis represents the position of SNPs in chromosome and the Y axis represents the −log_10_(*p*-value); The red dashed line represents a 10% genome-wide significance level (*p* = 1.79 × 10^−8^) after the Bonferroni correction; the green dashed line represents a 5% genome-wide significance level (*p* = 8.96 × 10^−9^) after the Bonferroni correction; and the blue dashed line represents a 1% genome-wide significance level (*p* = 1.79 × 10^−9^) after the Bonferroni correction. (**A**) Breast muscle weight; (**B**) Breast muscle percent; (**C**) Keel length; (**D**) Body weight; (**E**) Breast width; (**F**) Breast depth.

**Figure 5 genes-13-02153-f005:**
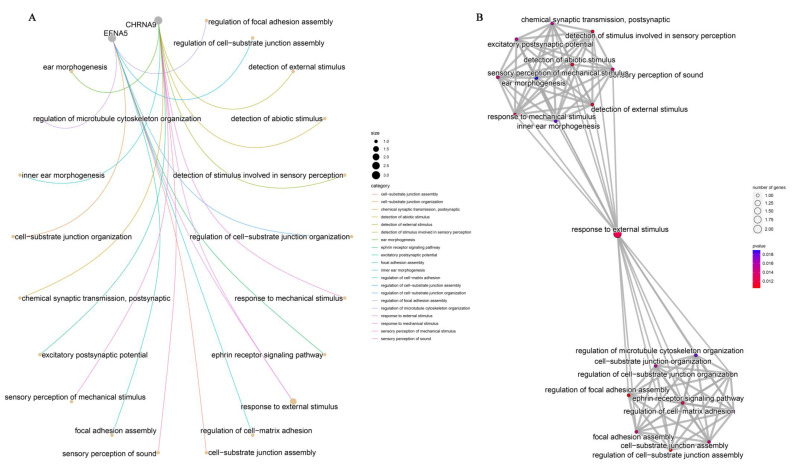
The GO annotation of candidate genes. (**A**) The GO annotation category. (**B**) The GO annotation category’s interaction.

**Figure 6 genes-13-02153-f006:**
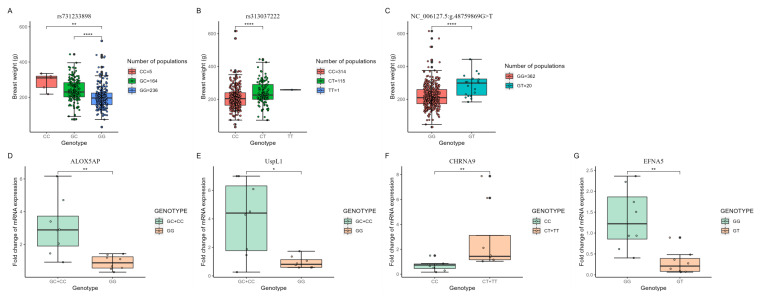
The breast muscle weight and expression levels of candidate genes in different genotypes. (**A**–**C**): The breast muscle weight of different genotypes. Vertical bars represent the mean ± standard error; (**D**–**G**): The expression levels of candidate genes in different genotypes. Vertical bars represent the mean ± standard error (n = 8). The scatter represents the sample distribution state around the mean. Statistically significant differences are indicated by * (*p* < 0.05), ** (*p* < 0.01), **** (*p* < 0.0001).

**Table 1 genes-13-02153-t001:** Dietary composition and nutrient levels.

Components	1–7 Weeks (%)	8–14 Weeks (%)
Corn	63.26	67.19
Soybean meal	30.2	18.88
Wheat bran	0.00	10.00
Fishmeal	2.50	0.00
Coarse powder	0.40	0.46
Fine stone powder	0.71	0.60
Dicalcium phosphate	1.50	1.50
Methionine	0.08	0.07
Salt	0.35	0.30
Metabolic energy	13.02	12.80
Protein	20.00	18.60
^a^ Commercial premix	1.00	1.00

^a^ Main components include vitamins (A, E, K, B1, B2, B6, and B12), trace elements (Cu, Zn, Mn, Fe, and Se), oxidative acid, calcium pantothenate, folate, biotin, and choline chloride.

**Table 2 genes-13-02153-t002:** The phenotypes and measurement methods.

Phenotypes	Measurement
Breast width	The straight-line distance on the body’s surface between two shoulder joints
Breast depth	The distance from the first thoracic vertebra to the anterior edge of the keel
Keel length	The distance from the front of the keel protrusion to the end
Full evisceration weight	The weight of the body after respiratory organs, digestive organs, reproductive organs, heart, abdominal fat, head, and feet removal
Breast muscle weight	The weight of the breast after skin and adherent fat removal
Sternum weight	The weight of sternum after muscle removal

**Table 3 genes-13-02153-t003:** qPCR primer information.

Gene	ID	Primer sequence (5′-3′)	Length (bp)	Produce Size (bp)	Annealing Temperature (°C)
*GADPH*	NM_204305.2	F: GACAGCCATTCCTCCACCTT R: AACTGAGCGGTGGTGAAGAG	20 20	222	59
*ALOX5AP*	NM_001278144.2	F: GGCAGAAGTACTTTGTGGGC R: TATGTGAAGCAGGCTGACCC	20 20	243	59
*USPL1*	NM_001162393.3	F: GGGAGAAAGGTCACTGATATCACAC R: AGAGGGAAGCAACACAAAAGTAAAC	20 25	281	59
*CHRNA9*	NM_204760.2	F: GCTGTTCACAGCCACGATGC R: TGCATCATACCAGCTTTGTCGAA	20 23	232	61
*EFNA5*	NM_001397572.1	F: CCAGATTCCAGCAGGGAGACT R: CCCACCTCTTGAACCCTTTGG	21 21	180	60

Note: All the primers are designed with Primer 6.0 software and synthesized by Beijing Qingke bioengineering company (Beijing, China). F, forward primer. R, reverse primer.

**Table 4 genes-13-02153-t004:** The traits of breast muscle development-related traits in the F2 population.

Trait	Sample Size	Max	Min	Mean	^a^ Sd	^b^ Cv (%)
Body weight (g)	478	3030.5	648.0	1569.05	363.25	23
Breast muscle weight (g)	478	616.0	73.0	213.92	70.25	33
Sternum weight (g)	478	231.0	43.5	90.42	30.16	33
Breast width (mm)	478	95.54	45.08	70.59	7.52	11
Breast depth (mm)	478	113.49	49.30	83.0	11.61	14
Keel length (mm)	478	165.21	63.46	98.11	9.85	10
Full evisceration weight (g)	478	1961	312	1054.15	267.92	25
^c^ Breast muscle percent (%)	478	31.41	7.10	20.19	3.53	18

Notes: ^a^ Standard deviation; ^b^ Coefficients of variation. The percentage of standard deviation in mean; ^c^ Percentage of breast muscle weight in the full evisceration weight.

**Table 5 genes-13-02153-t005:** The breast muscle development-related trait candidate SNPs and genes.

Trait ^a^	SNP ^b^	Pos (bp) ^c^	MAF ^d^	*p*-Value	−log10*p*	Allele	PVE (%) ^e^	Genes ^f^	Distance (bp) ^g^
Breast muscle weight	rs731233898	GGA1: 176412883	0.20	6.42 × 10^−10^	9.19	G/C	8.56	ALOX5AP USPL1	U:3688 U:16919
Breast percent	rs733077910	GGA1: 159574698	0.3	4.67 × 10^−12^	11.33	G/A	10.44	KLHL1 PCDH9	D:282679 D:868651
	rs738075929	GGA1: 159575031	0.31	8.26 × 10^−12^	11.08	G/C	8.37		
		GGA1: 159574275	0.29	9.54 × 10^−11^	10.02	A/G	9.22		
		GGA1: 73837197	0.26	8.26 × 10^−12^	12.03	A/G	10.33	KCNA1	D:38851
	rs315941028	GGAZ: 43140769	0.27	1.78 × 10^−12^	11.75	G/A	9.9	gag-pro	D:8
		GGA2: 80832197	0.23	7.74 × 10^−9^	8.11	A/G	7.49	Sun3 (LOC428505)	U:5539
Breast depth		GGA1: 73837197	0.26	6.7 × 10^−9^	8.17	A/G	9.73	KCNA1	D:12470
	rs733077910	GGA1: 159574698	0.3	8.3 × 10^−11^	10.08	G/A	10.44	KLHL1 PCDH9	D:282679 D:868651
	rs738075929	GGA1: 159575031	0.31	6.7 × 10^−9^	8.07	G/C	9.2		
		GGA1: 159574275	0.29	1.85 × 10^−10^	9.73	A/G	8.55		
		GGAW: 3138376	0.22	7 × 10^−11^	10.15	T/G	10.68	NEDD4L ERVK-8	Intronic:0 U:11796
		GGAW: 3138452	0.22	1.85 × 10^−10^	9.73	A/G	10.38		
	rs315941028	GGAZ: 43140769	0.27	4.77 × 10^−10^	9.32	G/A	10	gag-pro	D:8
		GGA2: 80832197	0.23	9.16 × 10^−9^	8.03	A/G	7.85	Sun3 (LOC428505)	U:5539
Body weight		GGA1: 73837197	0.26	9.16 × 10^−21^	20.87	A/G	20.7	KCNA1	D:12470
	rs733077910	GGA1: 159574698	0.3	1.77 × 10^−21^	20.75	G/A	21.45	KLHL1 PCDH9	D:282679 D:868651
	rs738075929	GGA1: 159575031	0.31	9.39 × 10^−17^	16.03	G/C	23.7		
		GGAW: 3138376	0.22	8.62 × 10^−17^	16.06	T/G	18.51	NEDD4L ERVK-8	Intronic:0 U:11796
	rs731233898	GGA1: 176412883	0.20	2.47 × 10^−10^	9.61	G/C	8.21	ALOX5AP Uspl1	U:3688 U:16919
	rs315941028	GGAZ: 43140769	0.27	1.60 × 10^−20^	19.79	G/A	23.23	gag-pro	D:8
		GGA2: 80832197	0.23	9.39 × 10^−19^	18.03	A/G	19.89	Sun3 (LOC428505)	U:5539
Keel length		GGA1: 73837197	0.26	1.9 × 10^−26^	25.72	A/G	21.82	KCNA1	D:12470
		GGA2: 80832197	0.23	5.25 × 10^−21^	20.28	A/G	20.25	Sun3 (LOC428505)	U:5539
		GGAZ: 48759869	0.02	9.52 × 10^−9^	8.02	G/T	5.32	EFNA5	D:424581
	rs733077910	GGA1: 159574698	0.3	3.58 × 10^−24^	23.45	G/A	23.21	KLHL1 PCDH9	D:282679 D:868651
	rs738075929	GGA1: 159575031	0.31	4.49 × 10^−25^	24.35	G/C	24.22		
		GGA1: 159574275	0.29	3.07 × 10^−24^	23.51	A/G	23.4		
	rs313037222	GGA4: 68943073	0.12	1.34 × 10^−8^	7.87	C/T	5.6	Rbm47 CHRNA9	D:9239 U:20141
	rs13973830	GGA1: 172204015	0.33	1.39 × 10^−8^	7.86	T/C	9.24	COG6	U:14052
	rs315941028	GGAZ: 43140769	0.27	1.19 × 10^−22^	21.93	G/A	24.29	gag-pro	D:8
		GGAW: 3138452	0.22	5.26 × 10^−21^	20.28	A/G	19.27	NEDD4L ERVK-8	Intronic:0 U:11796

^a^ Trait description in Table 3-1; ^b^ SNP rs ID from Ensembl; ^c^ Positions of the significant SNP according to the Gallus gallus. GRCg6a assembly Gallus gallus chromosome; ^d^ Minor allele frequency; ^e^ Phenotypic variation explained; ^f^ Gene located within 100 kb upstream and downstream of the significant SNP; ^g^ U/D represents the gene located upstream or downstream of the SNP (intergenic region).

## Data Availability

The variation data reported in this paper have been deposited in the Genome Variation Map (GVM) [65] in the National Genomics Data Center, Beijing Institute of Genomics, Chinese Academy of Sciences and China National Center for Bioinformation (https://ngdc.cncb.ac.cn/gvm, update on 28 October 2022), under accession number GVM000423.

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
