# Peer review of "Identification of New Genes and Genetic Variant Loci Associated with Breast Muscle Development in the Mini-Cobb F2 Chicken Population Using a Genome-Wide Association Study"

_genes, 2022, doi:10.3390/genes13112153_

Round 1

Reviewer 1 Report

In general, the document is well-written and of good quality. It is advisable that the authors improve or justify some details of the methodology. The conclusion should be improved

Author Response

"Please see the attachment

Reviewer 2 Report

The aim of this study was to explore the potential genetic mechanisms of breast muscle development in a F2 population constructed by reciprocal crosses between a fast-growing broiler hybrid (Cobb500) and a slow-growing native breed (Daweishan mini). The manuscript is well written but I would have a few small suggestions. Why was Spearman's correlation coefficient used, when it is used in the case of rank correlation. I think it would be more appropriate in this case to use the Pearson correlation coefficient. Line 199-200_explain in what sense the population is clearly differentiated. Additionally clarify why the variability of phenotypic values for breast muscle weight and sternum weight is highest.
